# Apelin, APJ, and ELABELA: Role in Placental Function, Pregnancy, and Foetal Development—An Overview

**DOI:** 10.3390/cells11010099

**Published:** 2021-12-29

**Authors:** Monika Dawid, Ewa Mlyczyńska, Małgorzata Jurek, Natalia Respekta, Karolina Pich, Patrycja Kurowska, Wiktoria Gieras, Tomasz Milewicz, Małgorzata Kotula-Balak, Agnieszka Rak

**Affiliations:** 1Laboratory of Physiology and Toxicology of Reproduction, Institute of Zoology and Biomedical Research, Jagiellonian University in Krakow, 30-387 Krakow, Poland; monika.dawid@doctoral.uj.edu.pl (M.D.); ewa.mlyczynska@doctoral.uj.edu.pl (E.M.); malgorzata99.jurek@student.uj.edu.pl (M.J.); natalia.respekta@doctoral.uj.edu.pl (N.R.); karolina.pich@doctoral.uj.edu.pl (K.P.); patrycja.kurowska@uj.edu.pl (P.K.); wiktoria.gieras@student.uj.edu.pl (W.G.); 2Department of Gynecological Endocrinology, Jagiellonian University Medical College, 31-501 Krakow, Poland; milewicz@interia.eu; 3University Centre of Veterinary Medicine JU-UA, University of Agriculture in Krakow, 30-059 Krakow, Poland; malgorzata.kotula-balak@urk.edu.pl

**Keywords:** adipokines, apelin, apelinergic system, placenta, pregnancy, pathology of pregnancy

## Abstract

The apelinergic system, which includes the apelin receptor (APJ) as well as its two specific ligands, namely apelin and ELABELA (ELA/APELA/Toddler), have been the subject of many recent studies due to their pleiotropic effects in humans and other animals. Expression of these factors has been investigated in numerous tissues and organs—for example, the lungs, heart, uterus, and ovary. Moreover, a number of studies have been devoted to understanding the role of apelin and the entire apelinergic system in the most important processes in the body, starting from early stages of human life with regulation of placental function and the proper course of pregnancy. Disturbances in the balance of placental processes such as proliferation, apoptosis, angiogenesis, or hormone secretion may lead to specific pregnancy pathologies; therefore, there is a great need to search for substances that would help in their early diagnosis or treatment. A number of studies have indicated that compounds of the apelinergic system could serve this purpose. Hence, in this review, we summarized the most important reports about the role of apelin and the entire apelinergic system in the regulation of placental physiology and pregnancy.

## 1. Introduction

Pregnancy is a dynamic period that includes a number of processes that are essential for the proper development of the foetus. Starting from fertilisation, through the implantation of the embryo in the uterine wall, then the development of unique structures such as the placenta, and finally the delivery itself, many changes occur in a woman’s body. They are conditioned by numerous processes taking place at the cellular level, such as proliferation, apoptosis, angiogenesis, or hormone secretion [1]. However, recent studies have indicated that myriad abnormalities in the function of these processes can cause pregnancy pathologies, including preeclampsia (PE), intrauterine growth restriction (IUGR), or gestational diabetes mellitus (GDM) [2]. The literature data indicate that the development of the above-mentioned disorders may contribute to the structural and functional dysfunction of the placenta itself, which during pregnancy is responsible for the connection of the mother’s and child’s circulatory systems, the formation of the placental barrier, and the secretion of numerous proteins and steroid hormones [3]. Due to the fact that the placenta seems to be the central organ for the exchange of substances between the maternal and foetal organism, knowledge about the substances involved in the regulation inside this organ seems to be invaluable in the context of searching for methods to improve the course of pregnancy—especially pregnancy complicated by specific disorders [4]. Research conducted in recent years has shown that such compounds can be found among white adipose tissue hormones; that is, adipokines. So far, many substances belonging to this group of compounds have been identified, and one of the better-known factors is apelin. Due to the large number of scientific reports on the expression and function of apelin and the entire apelinergic system, which also includes the apelin receptor (APJ) and its second endogenous ligand, ELABELA, in the body of humans and other animals, we aimed to review the knowledge of the role of apelin in the regulation of placental physiology and pregnancy. Moreover, in this review, we have collected data on apelin, APJ, and ELABELA, focusing on their expression and function in placental physiology, including proliferation, apoptosis, and hormone production during normal and pregnancy pathologies such as PE, IUGR, and GDM, as well as foetal development.

## 2. Apelin Structure, Expression, and Functions

Apelin is a biologically active protein that is produced mainly by white adipose tissue. This adipokine hormone was first isolated from bovine stomach extracts as an endogenous ligand of the previously identified APJ [5]. The gene that encodes apelin, *APLN*, is located on the long arm of the X chromosome at position Xq 25–26. The N-terminus of the protein contains the signal sequences and participates in the ligand–receptor interaction. The C-terminus plays a crucial role in maintaining the biological activity of the ligand [5,6]. There are many final forms of apelin, each of which comes from a common precursor, which is a 77-amino-acid pre-propeptide (Figure 1). After post-translational modification, pre-proapelin is transformed into endogenous isoforms such as apelin-36, apelin-17, apelin-13, apelin-16, or exogenous apelin-12. Interestingly, the above-mentioned variations of apelin differ from each other by the length of the polypeptide chain. Researchers have proven that the longer chains of this protein are characterised by lower biological activity, which is why they are converted into short-chain forms [7].

The most commonly used variant of apelin in laboratory tests is apelin-36, which has been found in the lungs, uterus, and testes [8]. The most active protein isoform is apelin-13, located in the mammary gland and hypothalamus [8]. Many studies have confirmed the important role of apelin in the human body, especially in the cardiovascular and reproductive systems (Figure 2). The regulatory effect of apelin has also been observed on the evolution and activity of the cardiovascular system; the apelin concentration was lower in patients with primary hypertension, ischaemic heart disease, or those who underwent infarction, compared with healthy individuals [9,10,11]. Other studies have shown that the increase in the level of apelin had a dual nature: it had a positive effect on the course of angiogenesis after ischaemic stroke [12], but it could also stimulate the process of cancer neoangiogenesis [13]. Recent research has shown that apelin could also promote proliferation of vascular smooth muscle cells (VSMC) [14] or rat ovarian granulosa cells (Gc) [15]. In addition, a number of studies indicated a significant role of apelin in the female reproductive system [16]. Its expression has been demonstrated, inter alia, in the ovaries of pigs, bovines, rhesus monkeys, and humans [16]. Studies conducted so far have indicated, for example, that apelin may regulate steroidogenesis in ovarian cells. Apelin significantly increased the secretion of progesterone (P4) and oestradiol (E2), and also increased the protein level of 3β—hydroxysteroid dehydrogenase/Δ5—4 isomerase (3βHSD) in human and porcine ovarian cells by activating the mitogen-activated protein kinase 3 (MAPK3) and 5’AMP-activated protein kinase (AMPK) pathways [17,18]. In turn, apelin is one of the factors that slows down apoptosis and increases proliferation in the ovaries by activating the protein kinase B (AKT) pathway [15].

Apelin also plays an important role in the treatment of carbohydrate disorders such as obesity or type II diabetes. When comparing the level of apelin in the plasma of people with morbid obesity and healthy people, the former group presented statistically significant hormone overproduction that occurred only in people with obesity as well as type II diabetes. Moreover, in people with type II diabetes, there was a correlation between the concentration of apelin, glucose, and triglycerides in the plasma [19]. Apelin also affects the course of cell proliferation, apoptosis, inflammatory processes, and angiogenesis. It has been shown that apelin inhibited pericyte apoptosis caused by hypoxia by reducing the expression of active caspase-3 and by increasing the Bcl-2/Bax ratio [20]. Moreover, apelin blocked the nuclear factor kappa light chain enhancer of the activated B-cell (NF-κB)/natural killer (NK) signalling pathway, which is responsible for inflammation, and thus reduced the production of proinflammatory cytokines [21].

## 3. APJ Structure, Expression, and Functions

We know that apelin acts through its specific receptor, APJ, encoded by the *APLNR* gene. The gene is located on chromosome 11 (q12), as determined by using fluorescent in situ hybridisation (FISH), and encodes a protein of 380 amino acids. Based on studies carried out in 1993 on human blood samples, APJ is a G-protein-coupled receptor (GPCR). The research has also confirmed its high (~50%) structural similarity to the angiotensin II receptor [22]. Moreover, previous studies have shown that the human APJ amino acid sequence was 92% homologous to that found in mice, and 90% homologous to that found rats [23,24]. In addition, there was 96% homology between the murine and rat sequences [24]. The APJ protein has seven hydrophobic transmembrane domains, which is a characteristic feature of proteins belonging to the GPCR family. Within the domains, there are sites for protein kinase A (PKA) phosphorylation, glycosylation, and palmitoylation [22]. The processes mentioned above are essential for the proper function of the receptor. The glycosylation of the N-terminus of GPCRs is responsible for the stability and expression of the receptor, proper protein folding, and binding to the ligand [25]. Moreover, palmitoylation of the C-terminus of GPCRs is important in the association of the receptor with the cell membrane, and the combination of this process with phosphorylation facilitates internalisation, dimerisation, and ligand attachment to GPCRs [26]. APJ messenger RNA (mRNA) expression has been demonstrated in mouse embryos, bovine follicles, and the central nervous system and peripheral tissues of humans and rats [22,23,24,27,28,29,30,31,32]. Studies have shown that insulin was a factor that increased the expression of APJ in adipose tissue [33]. In addition, APJ has two specific endogenous ligands, apelin and ELABELA (Table 1) [5,34]. It has been shown that apelin influenced the regulation of APJ expression in the gastrointestinal tract, and that the increased expression of APJ may be a consequence of repeated acute stress [35,36]. In addition, vascular endothelial growth factor (VEGF) and fibroblast growth factor (FGF) increase the expression of APJ and apelin in endothelial cells [37]. Schilffarth et al. [32] found that APJ, along with apelin, had an angiogenic effect, and affected the proliferation of capillaries; these changes mediated the selection of a preovulatory follicle, influencing the growth of the dominant follicle by increasing the supply of nutrients. The role of the apelin–APJ system in normal and pathological stages of pregnancy will be presented in Section 6 and Section 7. When discussing APJ, it is worth adding some information about its second endogenous ligand, namely ELABELA [34]. This peptide was first identified in 2013 from embryonic stem cells (ESC) in zebrafish [34,38]. The *APELA* gene encodes a pre-proprotein that consists of 54 amino acids in humans. The isoforms of ELABELA include ELA-32, ELA-21, and ELA-11. As a result of proteolysis, the ELABELA sequence is cleaved by furin, generating ELA-11 and ELA-21 [34]. However, cleavage of the signal peptide in the N-terminus produces a 32-amino-acid proprotein. ELA-32 is a mature form that, upon binding to APJ, becomes a biologically active molecule, just as other isoforms [34]. Yang et al. [39] observed a correlation between apelin (<0.2–0.6 nmol/L) and ELABELA (<0.2 to >0.6 nmol/L) concentration in human plasma. Interestingly, myriad data indicate that these ligands interacted differently with APJ. Moreover, physiological differences resulted from expression profiles and localisation. For example, between endothelial cells and fibroblasts, the expression levels of apelin and APJ were lower in fibroblasts, but the expression level of ELABELA was not significantly different in the two cell types [40]. Interestingly, human ESCs did not express APJ, which suggested these cells have another cell-surface receptor that can bind ELABELA [41]. Furthermore, the primary sequence, particularly on the C-terminus of ELABELA, is highly conserved in vertebrates. ELABELA itself is mostly expressed in ESCs, the vascular endothelium, the kidney, prostate tissue, and the human placenta [34]. Pauli et al. [38] showed that the ligand was responsible for self-renewal and regulation of apoptosis of ESCs. Moreover, this peptide is important in cell locomotion of gastrulation in zebrafish. In the same model, ELABELA—APJ influenced the development and formation of bone through modulating pluripotency factors in ventrolateral endodermal cells [42]. In addition, the decrease in ELABELA expression led to abnormalities in endoderm differentiation [43], impaired locomotion and cell differentiation during gastrulation, and serious cardiac dysplasia [34] in zebrafish. Many studies have demonstrated that ELABELA binding to APJ stimulated angiogenesis [44], regulated vasculogenesis [45], and decreased blood pressure [46] in adulthood. Interestingly, the level of ELABELA in plasma was correlated negatively with the degree of albuminuria in patients with noninsulin-dependent diabetes mellitus [47]. Furthermore, this endogenous ligand appears to play a crucial role in development, especially in the context of the cardiovascular system. It is worth adding that the previously mentioned differences between ELABELA and apelin are crucial in new therapeutic methods. Interestingly, Zheng et al. [48] showed decreased plasma levels of ELABELA in patients with high blood pressure. On the other hand, Sainsily et al. [49] administered high levels of salt, which induced hypertension and cardiorenal dysfunction in rats. In addition to lowering blood pressure, ELABELA had beneficial effects on other cardiovascular and renal dysfunctions through increased binding to APJ and improved resistance to apelin-13 cleavage enzymes of the renin–angiotensin–aldosterone system. Knockout of ELABELA or APJ led to cardiovascular disorders, which in turn increased mortality in mice [41]. As a future perspective, generating ELABELA analogues could allow successful treatment of cardiovascular diseases (e.g., congestive heart failure) and the regulation of water retention and hyponatremia [50,51].

## 4. Molecular Mechanism of Apelin Action

Apelin causes various effects in an organism due to the activation of different signalling pathways and the affinity of APJ to bind variants of this adipokine and to interact with G-protein isoforms (Gα, Gβ, and Gγ) (Figure 3) [5]. For example, apelin-13 could bind to the APJ Gi/o protein and inhibit the stimulatory effect of forskolin on 3′,5′—cyclic adenosine monophosphate (cAMP) in Chinese hamster ovary (CHO) cells [5]. Apelin could also bind APJ via Gαi2 and, consequently, activate the extracellular signal-regulated kinase 1/2 (ERK1/2) pathway in CHO and human embryonic kidney (HEK293) cells [52]. Interestingly, the effect of apelin is also tissue-dependent; Habata et al. [28] confirmed that apelin-13 and apelin-36 in CHO cells were not capable of generating calcium (Ca^2+^) mobilisation. On the other hand, in HEK293 cells and neurons, apelin-13 and apelin-36 did affect Ca^2+^ mobilisation [53]. Accumulated evidence indicates that apelin exerts its effect by activating multiple kinase pathways—for example, AKT phosphorylation by apelin occurred via a PTX-sensitive G-protein and protein kinase C (PKC) in human umbilical vein endothelial cells (HUVECs). This same model revealed that apelin induced the dual phosphorylation of the S6 ribosomal protein kinase (p70S6K) [54]. Of particular importance was the fact that activation of signalling pathway cascades was connected with different biological effects of this adipokine. Specifically, apelin stimulated proliferation by phosphoinositide 3-kinase (PI3K) phosphorylation in porcine ovarian follicles [17] while a similar effect was observed for apelin-13 in cultured rat cardiomyoblasts (H9c2) via activation of AKT and ERK1/2 [55]. Moreover, apelin protected mouse neurons from cell death by reducing reactive oxygen species (ROS) production and activation of actin kinase [56], while changes in AMPK phosphorylation by apelin-13 blocked the process of mouse neuronal apoptosis after stroke. A similar action was observed in retinas of mice, and the effect was reversed by APJ antagonist, and by inhibitors of the AKT and ERK1/2 signalling pathways [57]. Furthermore, in APJ knockout mice, apelin-13 negatively regulated AMPK by binding to APJ, with a consequent decrease in lipolysis, the hydrolytic degradation of triglycerides in adipose tissue to fatty acids and glycerol [58,59]. However, the aforementioned ELABELA also binds APJ and may activate G-protein and β-arrestin dependent pathways in the human heart. Apelin could increase cardiac contractility, ejection fraction, and cardiac output, and elicited vasodilatation in rats in vivo [39]. Moreover, referring to the large-scale research related to ELABELA in the cardiovascular system, which we mentioned in Section 3, this substance increased cardiac contractility in adult rat hearts in an ERK1/2-dependent manner [40]. Finally, in the absence of its ligands, APJ heterodimerised with other GPCRs and activated different signalling pathways [54].

## 5. Expression of Apelin, APJ, and ELABELA in the Placenta

The placenta is a crucial organ that provides a link between foetus and mother during pregnancy. The process of placentation begins after fertilisation, when the blastocyst adheres to the endometrium (inner layer of the uterus), which at this time is called the decidua, and the outer layer of cells (the trophoblast) starts to invade them (6–7 days postfertilisation (dpf) [60]. The trophoblast comprises two cell types that create inner and external layers called, sequentially, the cytotrophoblast and the syncytiotrophoblast [61,62] (Figure 4). The fusion between the cytotrophoblast and the syncytiotrophoblast is controlled by multiple factors such as cytokines, protein kinases, proteases, and transcription factors. At around 8–9 dpf, these placental layers form fluid-filled spaces, and at about 14 dpf, fluid-filled villus structures appear in the syncytiotrophoblast [63,64]. Before fusion, the cytotrophoblast appears similar to the syncytiotrophoblast, but the former comprises uninuclear cells derived from the trophectoderm, while the latter is multinucleated [62,65]. The syncytiotrophoblast has a mitotic index of 0, while the same parameter in the cytotrophoblast is about 1.5–2.9% [66,67]. The latest literature data suggest that the cytotrophoblast takes part in the metabolism of maternal-derived fatty acids and the biosynthesis of human placental lactogen (hPL) and human chorionic gonadotropin (hCG) [68,69]. During weeks 10–41 of pregnancy, the number of cells in the cytotrophoblast increases, but their volume is reduced. Moreover, in the late first trimester, the cytotrophoblast contains closely packed cells, but the structure is relatively thin [70,71]. Other studies suggested that during the first half of pregnancy, the syncytiotrophoblast was easy to identify due to a continuous row of nuclei, while in the second half of pregnancy, this structure became thinner and nuclei were located in groups, forming syncytial nuclear aggregates for better contact with placental blood vessels [72]. During pregnancy, the syncytiotrophoblast has the highest metabolic and endocrine activity in the placenta, and it participates in mother–foetus transfer of amino acids; water; gases; and hormones such as hCG, hPL, placental growth factor (PLGF), and leptin [60,73]. Moreover, the outer layer of the syncytiotrophoblast contains receptors for epidermal growth factor (EGF), insulin-like growth factor (IGF), and platelet-derived growth factor (PDGF) [74,75,76]. At 28 weeks of gestation, the syncytiotrophoblast surface is about 5 m^2^, while at full-term pregnancy, it reaches around 11–12 m^2^ [77]. During implantation, the developing syncytium invades the endometrium and forms fluid-filled spaces called lacunes (lacunar phase) [78]. About 3 weeks after fertilisation, the syncytiotrophoblast proliferates and is penetrated by the cytotrophoblast, after which the primary villi are created. Subsequently, secondary villi are formed by the growth of extraembryonic mesenchymal cells into the primary villi, and at 18 dpf, foetal capillaries penetrate these structures, creating tertiary villi [79]. After 3 months, the placenta is a fully developed organ that increases in size as a result of villus hypertrophy and elongation, reaching maturity at around 36–37 weeks of pregnancy [80].

Studies have shown that individual components of the apelinergic system are expressed in the tissue of placenta originating both from normal pregnancies and those complicated by specific disorders. Vaughan et al. [81] showed high expression of apelin in the human terminal placenta, almost 20 times higher than in adipose tissue; of note, its level did not differ between women with obesity and those with a normal body mass index (BMI). On the other hand, Cobellis et al. [82] documented that apelin expression decreased during the course of pregnancy in the cytotrophoblast and the syncytiotrophoblast, while APJ expression increased in the cytotrophoblast and endothelial cells. Similarly, in a rat model, the highest maternal plasma apelin levels occurred during the first trimester of pregnancy. Between middle and late pregnancy, there was an up to 50% decrease in apelin levels. The greatest reduction was observed in the last week of pregnancy, possibly due to elimination of apelin by the fetoplacental unit [83]. In support of these findings, lower serum apelin levels have been reported in pregnant women (24–28 weeks) compared with nonpregnant groups (4.45—8.7 ng/mL vs 5.0—9.3 ng/mL) [84]. Furthermore, Yamaleyeva et al. [85] observed that pyr-apelin-13 was the dominant form in this organ. Studies from our team have indicated higher apelin expression in JEG-3 placental cells, which reflect the cytotrophoblast, compared with BeWo cells, which reflect the syncytiotrophoblast; while APJ expression was the same in both cell lines [86]. These findings were in agreement with Cobellis et al. [82], who also observed higher apelin expression in the human cytotrophoblast. Moreover, immunohistochemistry of human placenta slides has shown high abundance of apelin in the cytoplasm of the endothelial lining of blood capillaries, as well as in maternal blood, and a moderate signal in placental arteries. In the case of APJ, a strong intensity was observed in syncytiotrophoblast cells [86]. As suggested by the authors of the above reports, apelin may affect the placentation process, as its expression was observed in particular in the cytotrophoblast, a highly proliferating layer, and may also be a key factor supporting the development of the foetus. It is worth noting that APJ and ELABELA are expressed at the very beginning of embryonic development, while the expression of apelin is observed at the end of gastrulation [34,87,88].

## 6. Effects of Apelin, APJ, and ELABELA on Placental Function

### 6.1. Proliferation

Proliferation of placental cells is regulated by cyclins, and their expression has been confirmed in various compartments of this organ [89]. Research has shown that in the human cytotrophoblast, expression of cyclins D3 and E decreases until delivery [90,91] Moreover, in the first trimester, cyclins A and B are expressed in the villous cytotrophoblast in the human placenta [92]. Danihel et al. [93] suggested that the cytotrophoblast was highly proliferative tissue, but the syncytiotrophoblast had limited proliferative properties. They also confirmed expression of proliferation markers proliferating cell nuclear antigen (PCNA) and Ki67 in the cytotrophoblast, mainly in the placenta during weeks 7–12 of pregnancy. In addition, Sun et al. [94] showed that cyclin G2 played a crucial role in the proliferation of human placental trophoblast cells. Recent studies have also shown that miR-518b promotes trophoblast cell proliferation in the human placenta via the Rap1b–Ras–MAPK pathway [95].

Using the human placental cell lines JEG-3 and BeWo, we proved that pyr-apelin-13 stimulated cell cycle progression, increasing the transition to the G2/M phase [86]. This was also visible in the altered cyclin expression profile: apelin stimulated especially the expression of cyclins D and E. There was increased proliferation of placental cells after stimulation with apelin at a concentration of 0.02–20 ng/mL, and APJ, ERK1/2/MAP3/1, signal transducer and activator of transcription 3 (STAT3), and AMPKα are involved in the molecular signalling pathways of this action (Figure 5) [86]. It is worth adding here that, unlike apelin, ELABELA may show an inhibitory effect on the proliferation of the trophoblast ex vivo, increasing its invasiveness, and the effect did not occur through interactions with APJ [96].

### 6.2. Apoptosis

Placental apoptosis is another key process for maintaining homeostasis, and many studies have shown the expression of apoptotic markers in the placenta. For example, Ka and Hunt [97] indicated the expression of inhibitor of apoptosis (IAP) in human cytotrophoblast cells, and in two cell lines, JEG-3 and JAR. In addition, Danihel et al. [93] demonstrated expression of the antiapoptotic protein Bcl-2 in the syncytiotrophoblast, and the proapoptotic protein Bax in both the cytotrophoblast and the syncytiotrophoblast of the human placenta during pregnancy; these findings confirmed the involvement of programmed cell death in this structure. Caspase-8 is important for syncytial fusion in the human trophoblast [98], and Bcl-2 is crucial for maintaining syncytial integrity in normal human pregnancy [99].

Apelin also has a positive effect on trophoblast survival by inhibiting the apoptosis of these cells. First, it reduces the mRNA expression of proapoptotic factors while it stimulates antiapoptotic factors at the mRNA and protein levels. In addition, it has the potential to reduce DNA fragmentation and the activation of effector caspase-3 and caspase-7 in BeWo cells. Importantly, this effect is more evident after induction of apoptosis with staurosporine, which could prove that apelin is able to prevent the effects of trophoblast damage, protect from harmful factors and, consequently, promote its survival. Here, the inhibition of effector caspases was also mediated by APJ and signal transduction of the ERK1/2/MAP3/1 and AKT pathways (Figure 5). The results observed in the BeWo cell line were supported by in vitro studies in human villous explants from the third trimester of pregnancy. Apelin upregulated the Bcl-2/Bax ratio, decreasing caspase-3 expression and DNA fragmentation. It also has the potential to reduce oxidative stress in BeWo cells. Oxidative stress is a major factor leading to accumulation of reactive oxygen species and, consequently, to cell death. As our research showed, apelin at a dose of 2 and 20 ng/mL may reduce oxygen metabolism efficacy and attenuate oxidative stress [100].

### 6.3. Endocrinology

During pregnancy, the placenta is a highly active endocrine organ that produces neuropeptides, pituitary-like hormones, adipokines, growth factors, steroid hormones, and adrenal-like peptides; the major source of these compounds is the syncytiotrophoblast [101]. One of them is hCG, which promotes differentiation of the cytotrophoblast into the syncytiotrophoblast by the luteinising hormone/choriogonadotropin receptor (LHCGR) and the PKA pathway [102]. In the primary human syncytiotrophoblast, hPL inhibits leptin production [103]. Cell-free hPL mRNA in maternal plasma may be associated with the abnormal invasiveness of the described organ [104]. The placenta is also a steroidogenic organ because it is crucial for foetal development, and steroid hormones (P4 and oestrogens) are produced in complex pathways involving mother, foetus, and placenta [105]. It is believed that in the human placenta, expression of cytochrome P450 family 17 subfamily A member 1 (CYP17A1) is insignificant, resulting in the inability to synthesise androgens [106], but a recent publication demonstrated CYP17A1 mRNA expression in human primary trophoblasts and in the JEG-3 and BeWo cell lines [107]. In addition, Hong et al. [108] suggested that compared with other species, in the human placenta, E2 had more pronounced effects on steroidogenesis than P4 via a positive feedback mechanism. The same researchers also showed that the expression of steroidogenic enzymes—CYP17A1, hydroxysteroid 17-beta dehydrogenase 3 (HSD17B3), and cytochrome P450 family 19 subfamily A member 1 (CYP19A1)—was elevated in the terminal stage of pregnancy, resulting in higher levels of E2 and dehydroepiandrosterone (DHEA).

The production and secretion of placental hormones that determine the proper course of gestation can be regulated by the apelinergic system. Previous studies indicated that the expression and secretion of apelin/ELABELA changed during various stages of pregnancy, which suggested that it may affect, inter alia, endocrine functions during this period [109]. Our previous research indicated that this adipokine may influence the endocrinology of pregnancy by regulating the secretion of human placental hormones. We have shown that apelin is able to reduce the secretion of trophoblast-derived steroid and protein hormones by blocking the expression of the steroidogenic enzymes 3βHSD and aromatase (CYP19), as well as protein hormones. Moreover, reduced secretion of PLGF and steroid hormones—that is, P4 and E2—occurs through APJ, PKA, and ERK1/2. In turn, reduced hCG, hPL, and PLGF secretion is only mediated by APJ and ERK1/2 (Figure 5) [110].

### 6.4. Angiogenesis

Angiogenesis, the growth of blood vessels, is the basis for better blood flow across the placenta [111]. Thanks to this process, the foetus develops in the correct conditions, taking into account all its metabolic requirements. The most important angiogenic factors are vascular endothelial growth factor (VEGF), FGF, and proteins belonging to the angiopoietin family (ANG) [112]. VEGF regulates vascular permeability, and is responsible for angiogenic processes in placental tissues of mice, sheep, and humans [113,114,115]. Moreover, in mice, VEGF knockout can cause defects in the angiogenesis and vasculogenesis of the placenta and foetus, leading to embryo mortality [116]. In addition to VEGF, another blood-flow-regulating factor is FGF, which is involved in increasing the proliferation of foetal and maternal arterial endothelial cells [112]. Interestingly, both VEGF and FGF in the vascular endothelium are involved in the production of nitric oxide (NO), which is one of the leading compounds involved in vasodilation [117]. However, in the case of proteins from the ANG family, their participation in angiogenic processes during normal development of the embryo is largely based on the regulation of endothelial cell survival and ensuring microvascular organisation [118,119]. Moreover, limitation in placental vessel development, and thus intensification of blood flow resistance in the vessels, may be the cause of embryo mortality [120,121].

Apelin reduces angiogenic activity during placental implantation, and hence contributes to the development of PE [122]. Moreover, other studies indicate that ELABELA–APJ has a significant role in vasculogenesis by the regulation of migration and differentiation of mesoendoderm cells during early embryonic development. In addition, apelin–APJ is strictly related to angiogenesis because of the important impact on endothelial cell proliferation and assembly during late embryonic development (Figure 5) [123].

### 6.5. Transport and Metabolism

When discussing placental angiogenesis, it is impossible to omit the processes of transporting certain substances, as well as their metabolism, during pregnancy. The placenta allows for the transport of many substances with biological activity and nutrients. For example, thyroid hormones that are crucial for foetal development are transported through the placenta. Chan et al. [124,125] showed expression of thyroid hormone transporters in the human placental cytotrophoblast, syncytiotrophoblast, and extra villous trophoblast. In the human placenta, aquaporins are expressed and may play a role in the regulation of amniotic fluid volume and transport of CO_2_, ammonia, and NO [126]. Another important compound crucial for steroidogenesis is cholesterol, which is transported by the placenta as very-low-, low- and high- density lipoprotein (VLDL, LDL, and HDL, respectively) from maternal circulation [127]. Moreover, around the second trimester of pregnancy, the developing foetus enlarges its dimensions several times, which is associated with a specific increase in the demand of its tissues for glucose. An increase in the transport of glucose through the placenta may occur through an intraplacental increase in the glucose concentration in response to a decrease in this parameter in the foetus relative to that in the mother’s body, or as a result of an improvement in the transport capacity of the placenta itself [128,129]. Furthermore, in the late stages of pregnancy, there are increased concentrations of triacylglycerol, phospholipids, and cholesterol in the mother’s plasma. During this time, the plasma levels of VLDL, LDL, and HDL increase, which are involved in the transport of long-chain polyunsaturated fatty acids (LC-PUFA) in the mother’s circulation [130]. On the other hand, the presence of receptors for individual lipoproteins in the placenta allows their uptake by the organ, where they are later hydrolysed by the following enzymes: lipoprotein lipase, phospholipase A2, and intracellular lipase. After this event, it is possible to release the fatty acids and metabolise them until they diffuse into the foetal plasma [130].

There are very few reports about apelin’s effects on placental transport or metabolic properties. It is known that this adipokine stimulates amino acid uptake in the human trophoblast. More precisely, pyr-apelin-13 at doses of 1–10.0 ng/mL increased System A amino acid transport, an important factor in foetal growth, but did not affect System L [81]. Moreover, apelin regulated glucose homeostasis in both the foetus and the newborn: intravenous administration of apelin to the mother increased glucose transport through the placenta, while intraperitoneal injection of adipokine in newborns increased the ability to uptake glucose in the lungs and muscles [131]. Moreover, during pregnancy, apelin is responsible for lipid metabolism, while ELABELA is involved in glucose metabolism [109].

In conclusion, proper placenta development, thus influencing the correct course of pregnancy, is conditioned by proper placental function, including several processes including proliferation, apoptosis, endocrinology, angiogenesis, and placental transport and metabolism. Apelin, by regulating the signalling pathways of numerous protein kinases, can be one of the factors that favourably regulates the present processes; it stimulates the proliferation and progression of the cell cycle through APJ and the signalling pathway of ERK1/2, STAT3, and AMPKα [86]; reduces placental cell apoptosis via APJ and kinases ERK1/2 and AKT [100]; and promotes the proper development of the placenta. Moreover, it has also been shown that apelin can modulate the secretion of placental protein hormones through ERK1/2 and steroid hormones through ERK1/2 and PKA, which in turn may be another decisive factor for the correct course of pregnancy [110]. Additionally, it has a beneficial effect on angiogenesis processes, which has an important impact on endothelial cell proliferation and assembly during late embryonic development [123]. Moreover, apelin has also been shown to play a significant role in the metabolism, transport, and maintenance of glucose homeostasis during pregnancy [109,131]. However, there is still a need for research to explain the molecular mechanisms of particular processes in different stages of pregnancy.

## 7. Placental Pathology and Pregnancy Pathology

Developmental defects of the placenta, abnormal placement of the placenta, and/or its failure represent serious threats to the course of pregnancy and foetal development. Placental development problems during early pregnancy lead most often to miscarriages. In advanced pregnancy, these problems are the cause of abnormal exchange between mother and foetus. Processes that occur incorrectly often lead to the development of numerous pregnancy pathologies that may inhibit the growth of the foetus and lead to other threats to the health and life of the mother and child. The most common disorders of pregnancy and the placenta are discussed below.

### 7.1. Preeclampsia

PE targets the liver, kidney, and brain, and is one of the major causes of increased maternal and foetal morbidity and mortality [132]. It is characterised by the development of arterial hypertension and proteinuria after 20 weeks gestation to 6 weeks postpartum in women who previously had normal blood pressure [133]. This disorder is associated with haematological dysfunction (e.g., thrombocytopaenia), nephrism, increased liver enzymes, and uteroplacental dysfunction [132]. The aetiology has not yet been fully explained; perhaps it is associated with abnormal development of the placenta in the first term and the maternal syndrome in the second and third terms, which is characterised by excess antiangiogenic factors. Besides, the occurrence of uteroplacental ischaemia leads to the release of proinflammatory cytokines into maternal circulation [134]. It is crucial to monitor perinatal pressure to provide early interventions and to reduce the risk of other complications.

Cobellis et al. [82] observed that during pregnancy in patients without PE, the immunohistochemical distribution of apelin decreased from the first to the third trimester of gestation in the cytotrophoblast, syncytiotrophoblast, and the stroma of placental villi. The APJ level increased in the cytoplasm of cytotrophoblast cells and in the cytoplasm of endothelial cells of normal placenta. In samples from women with PE, the authors observed that both apelin and APJ increased in all the placental compartments—the cytotrophoblast, the syncytiotrophoblast, and the stroma—with the greatest increase in the endothelial cells of the placental villi. On the other hand, Yamaleyeva et al. [85] observed that apelin content measured by radioimmunoassay (RIA) in human chorionic villi was lower in patients with PE compared with healthy pregnant women, while they did not observe differences in apelin mRNA expression and APJ mRNA and protein level. However, there are some findings that patients with PE had a higher serum level of apelin compared to healthy pregnant women of a similar chronological age, gestational age, and BMI, and the elevation of serum apelin levels might be a physiological feedback response to control the blood pressure in patients with PE [135]. Circulating apelin was significantly increased in early-onset PE, indicating the role of apelin in the discrimination of early-onset PE [136]. This observation was confirmed by Colcimen et al. [137] and correlated with upregulated VEGF, supporting the roles of haemodynamic alterations in foetoplacental circulation and structural alterations in uteroplacental tissue occurring in PE. Bortoff et al. [138] observed an opposite effect: the serum apelin concentration in patients with PE was lower than in control patients. This finding was confirmed by Taha et al. [139], as well as Mazloomi et al. [140] in pregnant women from Iran. Similarly, a lower serum level of apelin-13 was correlated strongly and inversely with systolic blood pressure, suggesting that in PE, the deterioration of the cardiovascular protective effect of apelin by other enzymes may also contribute to the deterioration of foetal development [141]. The maternal serum levels of apelin-36 and apelin-13 were significantly lower in patients with PE compared with control individuals [142]. Besides, apelin mRNA and protein were decreased in PE placenta, but were upregulated in maternal blood serum [122]. Interestingly, another study showed a low level of serum apelin-13 and the 1860 T > C polymorphism in the APLN gene promoter site, with increased allelic frequency of the CC genotype and the C allele compared with normotensive pregnant women; this factor might link cardiac complications in women with PE after delivery [143].

A rat model of PE, prepared by reducing the uterine perfusion pressure, was characterised by hypertension and poor pregnancy outcomes, such as decreased foetal and placental weight. Researchers treated the PE rats with apelin-13 at 6 × 10^−8^ mol/kg body weight twice a day; this treatment significantly ameliorated the symptoms of PE, improved the impaired endothelial nitric oxide synthase/NO signalling, and attenuated activation of oxidative stress in the rat model. They proposed that apelin may be a potential agent for preventing and treating PE [144]. In another study, researchers showed that the administration of pyr-apelin-13 at 2 mg/kg body weight/day reduced blood pressure and normalised the heart rate in a rat PE model compared with controls, and also normalised proteinuria in association with lower renal cortical collagen deposition [145]. Moreover, apelin ameliorated kidney damage in a PE rat model; hence, it might represent a curative candidate for prohibiting kidney damage [146].

It is interesting that ELABELA is secreted by the placenta and can participate in the development of PE. Ho et al. [41] showed that deletion of the APELA gene but not the APLN gene in mice caused PE symptoms such as kidney injury, hypertension, and proteinuria. Exogenous ELABELA infusion normalised hypertension, proteinuria, and birth weight. Panaitescu et al. [147] showed that plasma ELABELA levels were higher in patients with late-onset PE compared with normal pregnancy. They observed no differences between early-onset PE and normal pregnancies, similarly to Para’s research group [148]. However, Pritchard et al. [149] did not observe differences between placental mRNA expression of ELABELA and the circulating ELABELA level in serum of women with and without PE; these findings were confirmed by Ma et al. [48] in the first trimester of pregnancy. Deniz et al. [150] reported decreased ELABELA, apelin, and NO levels in the blood of pregnant women with PE (including severe PE) and in the venous arterial blood of newborns. Zhou et al. [151] obtained similar data for both the serum and placenta, indicating that future studies on the connection between ELABELA and PE are needed. Moreover, maternal blood ELABELA levels in the first and second trimesters were elevated in women who developed gestational hypertension late in pregnancy; these findings again highlighted the role of ELABELA in PE [152]. In a very recent report, ELABELA reversed the phenotypes of PE in mice and regulated the expression of mouse placental apoptosis factors by reducing the levels of apoptotic genes [153].

In conclusion, the different results between apelin/APJ/ELABELA expression in various compartments of the placenta and maternal plasma level in PE [85,135,136,137,138,139,140,141] may be explained by several factors such as: apelin corelation with different hormones/growth factors (e.g., VEGF, PLGF, IL-10 [154]), degree of intensity of PE (mild/severe cases) [155], or other environmental factors such as maternal age, smoking, or even BMI [138]. Based on the literature data, apelin inhibited the development of the rat model of PE; administration of apelin twice a day to rats significantly reduced the unfavorable symptoms of PE, which in turn was beneficial in repairing impaired endothelial nitric oxide synthase/NO signaling, and reduced the activation of oxidative stress in the rat model [144]. However, further research explaining molecular mechanism of apelin action on pregnancy pathology is needed.

### 7.2. Intrauterine Growth Restriction

IUGR leads to perinatal morbidity and growth impairment in childhood. The clinical definition of IUGR is an infant birth weight and/or length below the 10th percentile for the population at a given gestational age [156]. Genetic and environmental factors are the basis for the development of this disorder. Some risks are a young maternal age or smoking. A potential underlying cause is physiological remodelling of uterine spiral arteries. As a result, there is an abnormal nutrient supply, foetal hypoxia, and redistribution of blood to the most important organs of the foetus [134]. Moreover, infants with IUGR have been reported to have hypoglycaemia, hyponatraemia, respiratory distress syndrome, kidney diseases, metabolic diseases, necrotising enterocolitis, retinopathy of prematurity, and postnatal growth failure [157,158]. The symptoms after birth also include persistent pulmonary hypertension or pulmonary haemorrhage, respiratory distress, and glucose abnormalities [159].

There are some data about the apelinergic system in IUGR. Malamitsi-Puchner et al. [160] did not observe differences between IUGR cases and controls appropriate for gestational age, and there was a lack of correlation between apelin plasma concentration and IUGR. Of note, the foetuses had higher apelin concentrations than the mothers [160,161]. On the other hand, Van Mieghem et al. [162] showed that the apelin serum level was 30% lower in women with IUGR pregnancies compared with uncomplicated pregnancies; apelin mRNA expression in the placenta was similar for both groups. New research has also shown that maternal serum ELABELA was downregulated compared with control, and positively correlated with birth weight [163].

To sum up, the literature data on the apelinergic system in a complicated pregnancy with IUGR are quite insightful. However, the studies conducted so far indicated that significant changes in the levels of apelin/ELABELA have been observed in the placenta, maternal serum, and the foetus during IUGR pregnancy [157,158,159,160,161,162,163]. These studies suggested that manipulating the levels of apelinergic proteins may be important in restoring the normal course of pregnancy, thanks to which the foetal development will return to the right path. Unfortunately, these speculations should be confirmed in subsequent studies.

### 7.3. Gestational Diabetes Mellitus

GDM is a pathology that concerns 14% of pregnancies worldwide. During pregnancy, physiological insulin resistance (IR) develops, which facilitates the delivery of nutrients to the foetus. A slightly elevated glucose level stimulates the growth of the foetus [164]. However, pancreatic cell dysfunction leads to chronic IR. GDM is characterised by maternal hyperglycaemia and glucose intolerance. In addition, it increases the risk of obesity, noninsulin-dependent diabetes mellitus, and cardiovascular diseases in the long term [165]. GDM-linked carbohydrate intolerance detected in pregnancy is a risk for miscarriage, obesity, and cardiovascular diseases in adulthood [164].

It is unclear whether the apelin level changes in GDM: cross-sectional studies have reported unchanged and increased levels of this adipokine [166]. Besides, there was no link between apelin/APJ mRNA expression and GDM or the indices of IR [167]. Conversely, other data showed that the cord blood apelin level was significantly lower in women with GDM than control subjects, but no differences were observed in the maternal apelin level [168]. Besides, maternal serum apelin-36 levels were found to be higher in patients with GDM compared with control pregnant women. However, the cord blood apelin-36 levels were similar in the GDM and control groups. Moreover, maternal serum and cord blood apelin-36 levels correlated negatively with the gestational age and birth weight [169], data that indicated isoform-specific changes. Similarly, Mierzyński et al. [170] found no difference in apelin levels between patients with GDM and controls. In the most recent study, higher apelin and lower ELABELA levels were observed in patients with GDM compared with controls [109].

Overall, the studies also show significant changes in apelin/ELABELA levels during GDM pregnancy. Despite the lack of a clear relationship with the level changes described by the authors [109,166,167,168,169,170], the apelinergic system in GDM has different parameters compared to healthy pregnancies. Interestingly, it has been observed that the varied relationships in individual tissues can be directly related to specific isoforms of apelin [169]. To summarise, the apelin/ELABELA system appears to be related to the pathophysiological mechanisms of GDM, but further clinical evidence and experimental research are needed to elucidate these mechanisms.

Obesity is one of the factors that increase the risk of pregnancy complications. Moreover, adipokines are mediators of IR, and some adipokines are associated with the maternal metabolic state, which influences the transport of nutrients through the placenta [161]. Throughout pregnancy, there is a significant fluctuation in the secretion of various adipokines. Analysing and accounting for the role of adipokines should allow for a better understanding of the aetiology and pathophysiological mechanism of various gestational complications. Interestingly, PE, IUGR, and GDM are pregnancy pathologies linked in some way with the apelinergic system (Figure 6).

## 8. Perspectives

This review summarised the current reports on the apelinergic system in the development and function of the placenta. The collected results may have clinical application in the course of pregnancy and foetal development. Recent studies have shown the expression of APJ and its two specific ligands, namely apelin and ELABELA, in numerous tissues such as the lungs, uterus, testes, ovaries, and heart, where individual components show pleiotropic effects [8,16,48]. In addition, a number of studies indicated the presence of the apelinergic system in the placenta, where it participates in the regulation of the processes necessary for the proper course of pregnancy and proper development of the foetus: proliferation, apoptosis, endocrinology, angiogenesis, and the transport and metabolism of specific substances [81,86,100,110,122]. Recent studies indicated that disturbances in the above-mentioned processes may pose a serious threat to the course of pregnancy, thus affecting the morbidity and mortality of both mother and child [132]. The most common pathologies of pregnancy include PE, IUGR, and GDM, the consequences of which can be observed both in the foetal and postnatal periods [132,157,158,165]. Researchers have shown that the maternal serum level of apelin in patients with a pregnancy pathology increased or decreased compared with controls [136,138]. Similarly, a nonobvious result was obtained in the context of the plasma level of ELABELA in women with PE in relation to normal pregnancies [147]. Apelin serum levels were 30% lower in women with IUGR pregnancies compared with uncomplicated pregnancies, while the adipokine mRNA expression in the placenta was similar in both groups [163]. Moreover, in this pathology, the ELABELA level in maternal serum was downregulated compared with control and positively correlated with birth weight [162]. In the case of GDM, the obtained results were not unequivocal, indicating both an increase in the level of apelin and/or the apelinergic system, as well as a decrease or a lack of change in pregnancies complicated by this disorder [166]. In conclusion, apelin/ELABELA interacts through its receptor, and seems to be involved in the pathophysiological mechanism of PE, IUGR, and GDM, although further clinical evidence and studies must verify this mechanism.

## 9. Conclusions

In conclusion, this review has drawn attention to the importance of apelin and the entire apelinergic system in regulating the processes that determine the development of the placenta and the course of pregnancy. We have introduced the structure and function of the individual components of the mentioned system, namely apelin, ELABELA, and APJ. Moreover, we have drawn attention to the most important processes taking place in the placenta—proliferation, apoptosis, angiogenesis, hormone secretion, placental nutrient transport, and the metabolism of factors that determine the proper course of pregnancy—all of which influence the development of the foetus. In addition, we have taken into account the role of the apelinergic system in the most common pregnancy pathologies—PE, IUGR, and GDM—along with the determination of the molecular mechanisms of action of apelin and ELABELA in pregnancy. This summary of the knowledge about the described adipokines during pregnancy can be a fundamental basis for further research aimed at regulating the processes during pregnancy and preventing the aforementioned pathologies.

## Figures and Tables

**Figure 1 cells-11-00099-f001:**
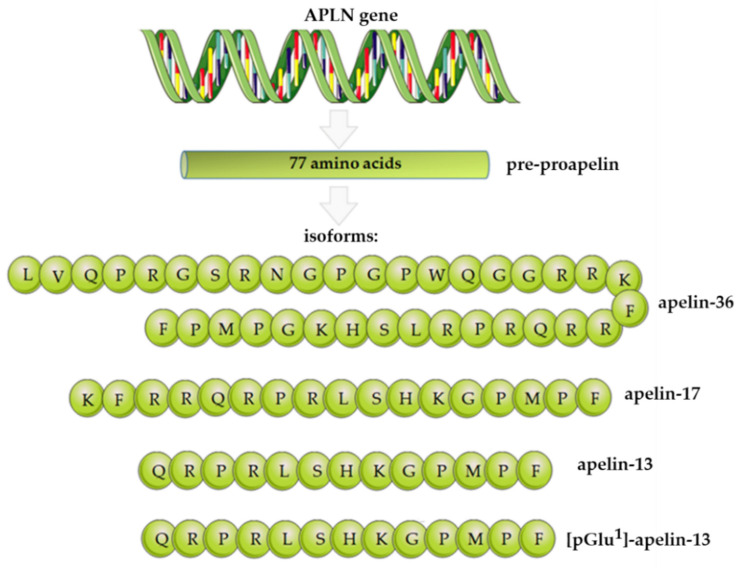
Apelin isoforms derived from the 77-amino-acid pre-propeptide. Based on Chen et al., 2003 [7].

**Figure 2 cells-11-00099-f002:**
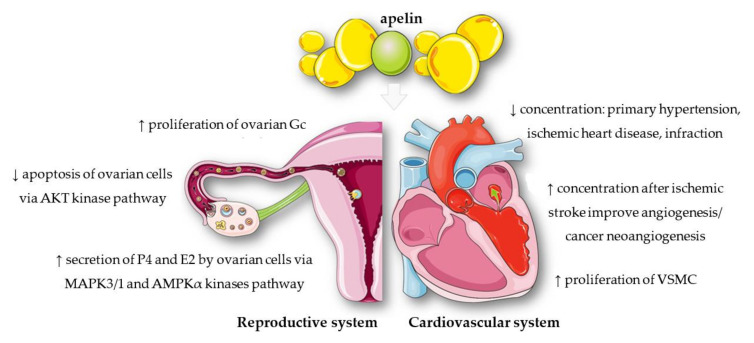
Role of apelin in reproductive and cardiovascular systems. MAPK3/1—mitogen-activated protein kinase 3/1; AMPKα—5’AMP-activated protein kinase; AKT—protein kinase B; VSMC—vascular smooth muscle cells; Gc—granulosa cells; E2—estradiol, P4—progesterone; ↑—increase; ↓—decrease.

**Figure 3 cells-11-00099-f003:**
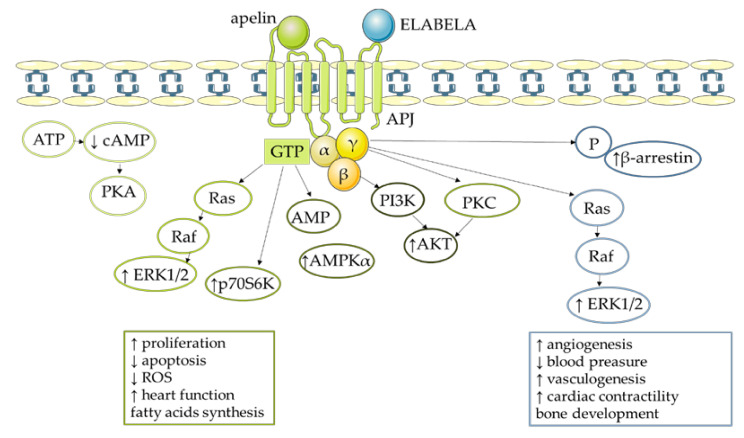
Activation of different signaling pathways and affinity of apelin/ELABELA to APJ receptor. AKT**—**protein kinase B; AMPKα**—**5’AMP**—**activated protein kinase; ERK1/2**—**extracellular signal activated kinase 1/2; P70s6k**—**ribosomal protein S6 kinase beta**—**1; PKC**—**protein kinase C; ATP**—**adenosine triphosphate; AMP**—**adenosine monophosphate; Camp**—**cyclic adenosine monophosphate; PI3K**—**phosphoinositide 3-kinase; ↑**—**increase; ↓**—**decrease.

**Figure 4 cells-11-00099-f004:**
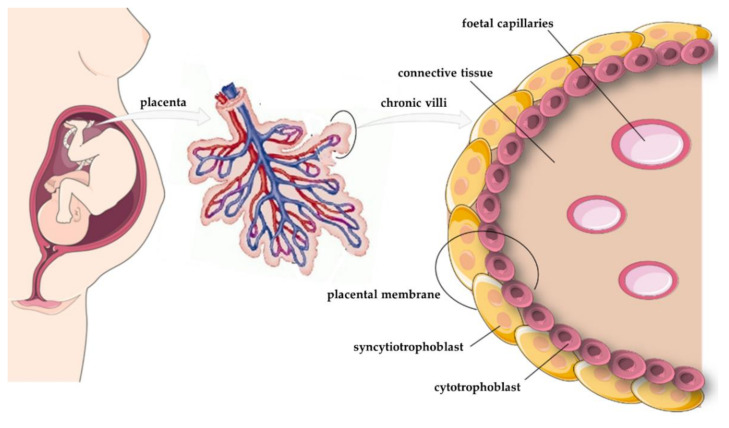
A simplified structure of the human placenta, taking into account the cross-section through the placental villi.

**Figure 5 cells-11-00099-f005:**
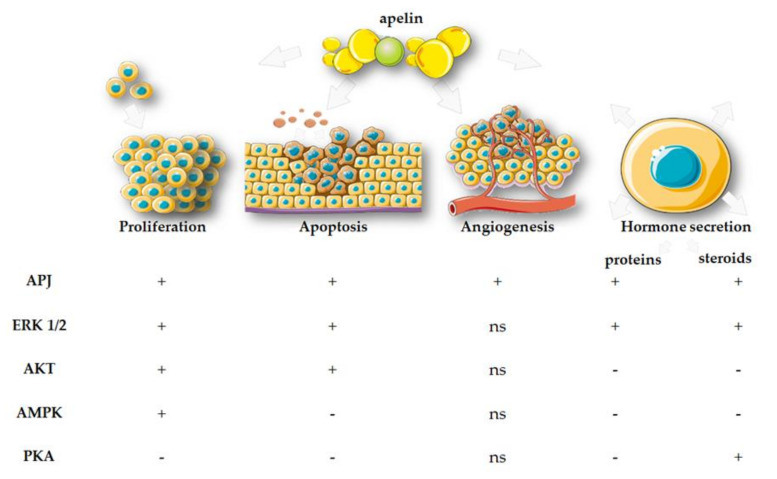
Signal molecular pathways of placental processes regulated by apelin. APJ**—**apelin receptor; ERK 1/2**—**extracellular signal activated kinase 1/2; AKT**—**protein kinase B, AMPK**—**5’AMP-activated protein kinase; PKA**—**protein kinase A; +/– **—**occurs/does not occur through; ns**—**no study.

**Figure 6 cells-11-00099-f006:**
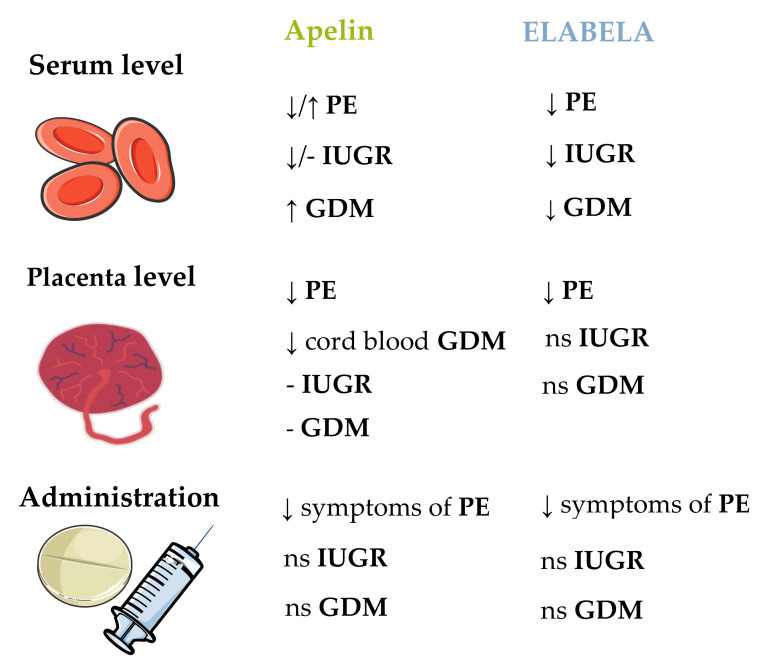
Changes in serum and placental apelin levels, as well as its administration effect during pregnancy pathologies. PE**—**preeclampsia; IUGR**—**intra uterine growth restriction; GDM**—**gestational diabetes mellitus; ↑**—**increase; ↓**—**decrease; ns**—**no study.

**Table 1 cells-11-00099-t001:** Apelinergic system: characteristic and function of mail compounds. Gc**—**granulosa cells; VSMC**—**vascular smooth muscle cells; ESC**—**embryonic stem cells; ↑**—**increase; ↓**—**decrease.

Apelinergic System	Apelin	ELABELA	APJ
Approved gene symbol	APLN	APELA	APLNR
Approved name	Apelin	Apelin receptor early endogenous ligand	Apelin receptor
Chromosomal location	Xq26.1	4q32.3	11q12.1
Gene groups	Neuropeptides	Receptor ligands	Receptors
First isolation	Bovine stomach extracts	ESC in zebrafish	Human blood
Function in organisms	↑ Carbohydrate disorders treatment ↓ Pericytes apoptosis ↑ Proliferation of Gc/VSMC ↓ Proinflammatory cytokines production ↑ Cancer neoangiogenesis	↓ Gastrulation disorders ↑ Blood vessel angiogenesis/vasculogenesis ↓ Renal dysfunctions ↓ Blood pressure ↓ Cardiovascular disorders	↑ Signal transmission pathway from apelin/ELABELA

## Data Availability

Not applicable.

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
