# Peer review of "Apelin, APJ, and ELABELA: Role in Placental Function, Pregnancy, and Foetal Development—An Overview"

_cells, 2021, doi:10.3390/cells11010099_

Round 1
Reviewer 1 Report
In their review, the Authors have performed an extraordinary effort in summarizing the current evidence on the importance of apelin and the entire apelinergic system in regulating the processes that determine the development of the placenta and the course of pregnancy. In addition, they have taken into account the role of the apelinergic system in the most common pregnancy pathologies –PE, IUGR and GDM – along with the determination of the molecular mechanisms of action of apelin and ELABELA in pregnancy. This special effort is a fundamental basis for further research aimed at regulating the processes during pregnancy and preventing the principal pathologies.
The readability is good, references are appropriate and updated. In the opinion of this reviewer, the paper might be useful for knowledge in earlier stage of education in obstetrics and a god start point for investigations, mainly for open questions emerging during its lecture.
Author Response
We thank very much Reviewer 1 for his/her thorough review and comments.
We thank very much Reviewer 1 for his/her thorough review and comments.
Reviewer 2 Report
This article is an interesting review of Apelin, APJ, ELABELA. This review is, however, insufficient to draw the conclusion that Apelin is an important substance during pregnancy. In your paper, it is difficult to determine whether apelin and related substrates are involved in the development of IUGR, PR, GDM.Clear data should be shown those are involved in the onset of these gestational complications. However, there is no clear explanation that these are the important substances.
The inconsistent difference in blood levels of Apelin and related substrates between the PE, GDM, or IUGR and the normal might indicate that it is simply the result of PE and GDM.
Table 2 merely shows the characteristics of those disease and does not show the association of Apelin with the complications. Therefore Table 2 should be excluded. Simple explanations for them are dominated and recurring. These should be briefly described. In contrast, the involvement of Apelins in pregnancy complications has not been specifically stated. What we are expecting in this article is sparse. ..
You said that Apelins play important roles in the development and function of the placenta, but it is unclear how it is involved in the process. Based on the above, this article should be rewritten as a paper that clarifies that the apelin metabolic system is involved in the onset of pregnancy complications. I'm looking forward to a good rewritten article
Author Response
We thank very much Reviewer 2 for his/her thorough review and comments. Please find below our response to concerns regarding the present work.
This article is an interesting review of Apelin, APJ, ELABELA. This review is, however, insufficient to draw the conclusion that Apelin is an important substance during pregnancy. In your paper, it is difficult to determine whether apelin and related substrates are involved in the development of IUGR, PR, GDM. Clear data should be shown those are involved in the onset of these gestational complications. However, there is no clear explanation that these are the important substances. The inconsistent difference in blood levels of Apelin and related substrates between the PE, GDM, or IUGR and the normal might indicate that it is simply the result of PE and GDM.
We are agree with Reviewer 2, it is difficult to discuss whether apelin and related substrates are involved in the development of IUGR, PR, GDM because there are limitation of exist literature. However, there are some studies which may explain involvement of apelinergic system only in PE. On page 14 (lines 552-561) we add a summary about available data. Differences between apelin/APJ/ELABELA expression in various compartments of the placenta and maternal plasma level in PE [86, 136-142] may be explain by several factors like: apelin corelation with different hormones/growth factors e.g. VEGF, PLGF, IL-10 [155], degree of intensity of PE (mild/ severe cases) [156] or other environmental factors like maternal age, smoking or even BMI [139]. Some literature data documented that administration of apelin to rats model PE significantly ameliorated the symptoms of PE, improved the impaired endothelial nitric oxide synthase/NO signalling and attenuated activation of oxidative stress in rat model and proposed that apelin may be a potential agent for preventing and treating PE. However we are aware that further research is needed to confirm these relationships.
Conclusion section summarizing apelinergic system levels in placenta tissue, plasma level or effect of PE development was added after PE, GDM, IUGR description, as suggestion.
Table 2 merely shows the characteristics of those disease and does not show the association of Apelin with the complications. Therefore Table 2 should be excluded. Simple explanations for them are dominated and recurring. These should be briefly described. In contrast, the involvement of Apelins in pregnancy complications has not been specifically stated. What we are expecting in this article is sparse.
Table 2 was removed an more explain was added in text, as a suggestion.
You said that Apelins play important roles in the development and function of the placenta, but it is unclear how it is involved in the process. Based on the above, this article should be rewritten as a paper that clarifies that the apelin metabolic system is involved in the onset of pregnancy complications. I'm looking forward to a good rewritten article.
Thank you for this important comment, we completed effect of apelinergic system on placenta development. After entire section no 6 – page 12 (lines 450 – 466) we add a necessary summary of the available data. We sum up that apelin by regulating the signaling pathways of numerous protein kinases, for example, may be one of the factors that favorably regulate the present processes. It stimulates the proliferation and progression of the cell cycle through the signaling pathway of APJ, ERK1 / 2, STAT3 and AMPKα [87] and reduces the process of human placental cell apoptosis by phosphorylation of APJ, ERK1 / 2, AKT [101], which promotes the proper development of tissue structures. Moreover, it has also been shown that the described adipokine can modulate the secretion of placental protein hormones through ERK1 / 2 and steroid hormones through ERK1 / 2 and PKA, which in turn may be another decisive factor for the correct course of pregnancy [111]. Additionally, it has a beneficial effect on angiogenesis processes, which has the important impact on endothelial cell proliferation and assembly during late embryonic development [124]. Moreover, apelin has also been shown to play a significant role in the metabolism, transport and maintenance of glucose homeostasis during pregnancy [110,132]. However there is still a need for research to explain the mechanisms of particular processes in different stages of pregnancy.
We would like to thank Reviewer for time and effort reviewing this manuscript. I believe that the valuable hints will raised the value of this scientific publication. I hope that introduced changes solved the problem raised.
Reviewer 3 Report
This review provides a lot of information even if it is not original
-Please improve the quality of figure 6
Author Response
We thank very much Reviewer 3 for his/her thorough review and comments. Quality of Figure 6 was improve, as suggestion.
Reviewer 4 Report
Comments for authors:
Title:
Apelin, APJ and ELABELA: Role in Placental Function, Pregnancy and Foetal Development – An Overview
Authors:
Dawid M., Mlyczynska E., Jurek M., Respekta N., Pich K., Kurowska P., Gieras W., Milewicz T., Kotula-Balak M., Rak A.
Manuscript ID: cells-1459363
Objective:
In the current review, the authors highlight the effects of the apelin system, with the two specific ligands apelin and ELABELA, on placental function, fetal development, including pathological events such as preeclampsia and GDM.
Page 5 – lines 199-200:
“Sainsily et al. [49] administered high levels of salt to patients.”
Do the authors mean rats – instead of patients? This paper reports about SHR, i.e., spontaneously hypertensive rats.
Page 13 – line 506-511:
In the first sentence, the authors describe that “Yamaleyeva et al. [87] observed lower apelin content in patients with PE compared with control patients. In the next sentence they mention, that “these are the first findings that patients with PE have a higher serum level of apelin than healthy patients. This is a contradiction that must be clarified. Furthermore, the term “patient” is inappropriate for control and healthy subjects.
Page 13 – line 512:
Ref. 138 is Kucur et al. instead of Kocur – this needs to be improved.
Page 14 – line 523:
The maternal – instead of “ma-ternal”
Page 14 – line 529:
… might link cardiac complications – instead of “complica-tions”
Page 14 – line 540:
in association … - instead of “associa-tion”
Page 14 – line 541:
Kidney instead of “kid-ney”
Page 14 – line 546:
Exogenous instead of “Exog-enous”
Page 14 – line 549:
Early-onset instead of “ear-ly-onset”
Page 14 – line 551:
… circulating … instead of “circulat-ing”
Page 15 – line 579:
IUGR - instead of “IGUR”
Figures 1 and 2 were black at the edges when printed.
References:
Ref. 2 – Journal:
2021, 21, 100503 – instead of “10050”
Ref. 5 – Authors:
… ZOU M.X., Kawamata Y., Fukusumi S., Hinuma S., Kitada C., Kurokawa T., Onda H., Fujino M. – better et al.
Ref. 6 – Authors:
O´Carrol A.-M., Lolait S.L., Harris L.E., Pope G.R.
Ref. 7 – Authors:
Chen M.M., Ashley E.A., Deng D.X., Tsalenko A., Deng A., Tabibiazar R., Ben-Dor A., Fenster B., Yang E., King J.Y. et al.
Ref. 9 – Authors:
Weir R.A.P., Chong K.S., Dalzell J.R., Petrie C.J., Murphy C.A., Steedman T., Mark P.B., McDonagh T.A., Dargie H.J., McMurray J.J.V.
Ref. 10 – Authors:
Kuklinska A.M., Sobkowicz B., Sawicki R., Musial W.J., Waszkiewicz E., Bolinska S., Malyszko J.
Ref. 11 – Authors:
Sonmez A., Celebi G., Erdem G., Tapan S., Genc H., Tasci I., Ercin C.N., Dogru T.,Kilic S., Uckaya G. et al.
Ref. 12 – Authors:
Chen D., Lee J., Gu X., Wei L., Yu S.P.
Ref. 13 – Authors and Journal:
Sorli S.C., Le Gonidec S., Knibiehler B., Audigier Y.
And
2007, 26, 7692-7699 – instead of „2008“
Ref. 14 – Journal:
Front Biosci - instead of Front “Bio Sci”
Ref. 16 – Journal:
2018, 2018, 9170480 – instead of 2018, “6”, 9170480
Ref. 19 – Authors:
Garcia-Almeida J.M.
And
Tinahones F.J.
Ref. 20 – Authors and Journal:
Jiang Y.R. and
2015, 2015, 186946
Ref. 22 – Authors:
… Kennedy J.L., Shi X., Petronis A., George S.R., Nguyen T.
Ref. 25 – Journal:
Hum Reprod Update – instead of Hum Reprod “update”
Ref. 32 – Authors:
MEYER, H.”H.D.”
Ref. 35 – Authors:
O´Carroll A.-M. - instead of “A.M.”
Ref. 36 – Authors:
GREELEY Jr. G.H.
Ref. 38 – Authors:
CHEW G.-L.
Ref. 40 – Authors:
KILPIÖ T. – instead of “Kilpio” T.
Ref. 41 – Authors:
GOH G.H.-Y.
Ref. 42 – Authors and Journal:
TEH C.
2016, 5, e11475
Ref. 43 – Authors and Title:
FORD G.H.;
… Elabela- and apj-deficient hearts.
Ref. 45 - Journal:
2015, 4, e06726
Ref. 46 - Authors:
LONGPRE, J-M.
Ref. 47 - Title:
Serum Elabela/toddler – instead of “SerumElabela/toddler” …
Ref. 48 – Journal:
Front Cardiovasc. Med.
Ref. 49 - Authors:
Longpre J.-M.
Ref. 50: Authors:
HSU C-W; and et al.
Ref. 51 - Citation:
2018, 17, 169
Ref. 53 - Title:
co-receptor
Ref. 56 – Journal:
2018, 17, 447-451
Ref. 60 – Authors:
ZHANG X.-J. and et al.
Re. 61 – Authors:
KILPIÖ T.
Ref. 65 – Authors:
PÖTGENS A.J.G. and FRANK H.-G.
Ref. 67 – Authors, Year:
ARNHOLDT H., DIEBOLD J., KUHLMANN B., LÖHRS U.
Year 1991
Ref. 77 – Authors:
CHEAH F.C. and TAN A.E.
Ref. 78 – Authors:
… van Zuylen, W.J., SCOTT G.M., RAWLINSON W.D.
Ref. 86 – Journal:
Clin Endocrinol (Oxf) 2011, 75, 367-371
Ref. 90 – Authors:
ZENG X.-X. I.
Ref. 97 – Authors:
LI Y.-X. and WANG Y.-L.
Ref. 98 – Title:
… major source of …
Ref. 100 – Authors:
EMANS N. instead of “Means” N.
Ref. 111 – Authors:
GUO Y.-Y., … LI Y.-C., HU H.-T., SU Y.-F. … Wang Y.-Y.
Ref. 116 – Journal:
J Reprod Fertil
Ref. 124 – Journal:
Hypertens Pregnancy
Ref. 126 – Authors:
CHAN S.-Y.
Ref. 130 – Journal:
Am J Physiol
Ref. 131 – Journal:
PMPH-USA Limited. - Space –
Ref. 133 – Authors:
… WATTEZ J.-S., LUKASZESKI M.-A.,…
Ref. 141 – Authors:
TAHA A. S. instead of Taha S. A.
Ref. 143 – Authors:
TEMUR, instead of Temur”.”
Ref. 148 – Authors:
Diab A.A.A.
Ref. 150 – Journal:
Are the authors feels certain with volume 11?
HSU C.-D.
Ref. 156 – Journal:
2009, 3, 332-336 - instead of 2009, “6”, 332-336
Ref. 157 – Author and Journal:
DÖTSCH J. – instead of “Dotsch” J.
2014, 1, 2
Ref. 170 – Journal.
2021, 2021, 5547228 – instead of 2021, “9”, 5547228
Author Response
We thank very much Reviewer 4 for his/her thorough review and comments. Please find below our response to concerns regarding the present work.
Title: Apelin, APJ and ELABELA: Role in Placental Function, Pregnancy and Foetal Development – An Overview
Authors: Dawid M., Mlyczynska E., Jurek M., Respekta N., Pich K., Kurowska P., Gieras W., Milewicz T., Kotula-Balak M., Rak A.
Objective: In the current review, the authors highlight the effects of the apelin system, with the two specific ligands apelin and ELABELA, on placental function, fetal development, including pathological events such as preeclampsia and GDM.
Individual points were corrected according to the suggestions proposed by the Reviewer 4.
Page 5 – lines 199-200:
“Sainsily et al. [49] administered high levels of salt to patients.”
Do the authors mean rats – instead of patients? This paper reports about SHR, i.e., spontaneously hypertensive rats.
Page 13 – line 506-511:
In the first sentence, the authors describe that “Yamaleyeva et al. [87] observed lower apelin content in patients with PE compared with control patients. In the next sentence they mention, that “these are the first findings that patients with PE have a higher serum level of apelin than healthy patients. This is a contradiction that must be clarified. Furthermore, the term “patient” is inappropriate for control and healthy subjects
Page 13 – line 512:
Ref. 138 is Kucur et al. instead of Kocur – this needs to be improved.--> byÅ‚o et at.
Page 14 – line 523:
The maternal – instead of “ma-ternal”
Page 14 – line 529:
… might link cardiac complications – instead of “complica-tions”
Page 14 – line 540:
in association … - instead of “associa-tion”
Page 14 – line 541:
Kidney instead of “kid-ney”
Page 14 – line 546:
Exogenous instead of “Exog-enous”
Page 14 – line 549:
Early-onset instead of “ear-ly-onset”
Page 14 – line 551:
… circulating … instead of “circulat-ing”
Page 15 – line 579:
IUGR - instead of “IGUR”
Figures 1 and 2 were black at the edges when printed.
References:
Ref. 2 – Journal:
2021, 21, 100503 – instead of “10050”
Ref. 5 – Authors:
… ZOU M.X., Kawamata Y., Fukusumi S., Hinuma S., Kitada C., Kurokawa T., Onda H., Fujino M. – better et al.
Ref. 6 – Authors:
O´Carrol A.-M., Lolait S.L., Harris L.E., Pope G.R.
Ref. 7 – Authors:
Chen M.M., Ashley E.A., Deng D.X., Tsalenko A., Deng A., Tabibiazar R., Ben-Dor A., Fenster B., Yang E., King J.Y. et al.
Ref. 9 – Authors:
Weir R.A.P., Chong K.S., Dalzell J.R., Petrie C.J., Murphy C.A., Steedman T., Mark P.B., McDonagh T.A., Dargie H.J., McMurray J.J.V.
Ref. 10 – Authors:
Kuklinska A.M., Sobkowicz B., Sawicki R., Musial W.J., Waszkiewicz E., Bolinska S., Malyszko J.
Ref. 11 – Authors:
Sonmez A., Celebi G., Erdem G., Tapan S., Genc H., Tasci I., Ercin C.N., Dogru T.,Kilic S., Uckaya G. et al.
Ref. 12 – Authors:
Chen D., Lee J., Gu X., Wei L., Yu S.P.
Ref. 13 – Authors and Journal:
Sorli S.C., Le Gonidec S., Knibiehler B., Audigier Y.
And
2007, 26, 7692-7699 – instead of „2008“
Ref. 14 – Journal:
Front Biosci - instead of Front “Bio Sci”
Ref. 16 – Journal:
2018, 2018, 9170480 – instead of 2018, “6”, 9170480
Ref. 19 – Authors:
Garcia-Almeida J.M.
And
Tinahones F.J.
Ref. 20 – Authors and Journal:
Jiang Y.R. and
2015, 2015, 186946
Ref. 22 – Authors:
… Kennedy J.L., Shi X., Petronis A., George S.R., Nguyen T.
Ref. 25 – Journal:
Hum Reprod Update – instead of Hum Reprod “update”
Ref. 32 – Authors:
MEYER, H.”H.D.”
Ref. 35 – Authors:
O´Carroll A.-M. - instead of “A.M.”
Ref. 36 – Authors:
GREELEY Jr. G.H.
Ref. 38 – Authors:
CHEW G.-L.
Ref. 40 – Authors:
KILPIÖ T. – instead of “Kilpio” T.
Ref. 41 – Authors:
GOH G.H.-Y.
Ref. 42 – Authors and Journal:
TEH C.
2016, 5, e11475
Ref. 43 – Authors and Title:
FORD G.H.;
… Elabela- and apj-deficient hearts.
Ref. 45 - Journal:
2015, 4, e06726
Ref. 46 - Authors:
LONGPRE, J-M.
Ref. 47 - Title:
Serum Elabela/toddler – instead of “SerumElabela/toddler” …
Ref. 48 – Journal:
Front Cardiovasc. Med.
Ref. 49 - Authors:
Longpre J.-M.
Ref. 50: Authors:
HSU C-W; and et al.
Ref. 51 - Citation:
2018, 17, 169
Ref. 53 - Title:
co-receptor
Ref. 56 – Journal:
2018, 17, 447-451
Ref. 60 – Authors:
ZHANG X.-J. and et al.
Re. 61 – Authors:
KILPIÖ T.
Ref. 65 – Authors:
PÖTGENS A.J.G. and FRANK H.-G.
Ref. 67 – Authors, Year:
ARNHOLDT H., DIEBOLD J., KUHLMANN B., LÖHRS U.
Year 1991
Ref. 77 – Authors:
CHEAH F.C. and TAN A.E.
Ref. 78 – Authors:
… van Zuylen, W.J., SCOTT G.M., RAWLINSON W.D.
Ref. 86 – Journal:
Clin Endocrinol (Oxf) 2011, 75, 367-371
Ref. 90 – Authors:
ZENG X.-X. I.
Ref. 97 – Authors:
LI Y.-X. and WANG Y.-L.
Ref. 98 – Title:
… major source of …
Ref. 100 – Authors:
EMANS N. instead of “Means” N.
Ref. 111 – Authors:
GUO Y.-Y., … LI Y.-C., HU H.-T., SU Y.-F. … Wang Y.-Y.
Ref. 116 – Journal:
J Reprod Fertil
Ref. 124 – Journal:
Hypertens Pregnancy
Ref. 126 – Authors:
CHAN S.-Y.
Ref. 130 – Journal:
Am J Physiol
Ref. 131 – Journal:
PMPH-USA Limited. - Space –
Ref. 133 – Authors:
… WATTEZ J.-S., LUKASZESKI M.-A.,…
Ref. 141 – Authors:
TAHA A. S. instead of Taha S. A.
Ref. 143 – Authors:
TEMUR, instead of Temur”.”
Ref. 148 – Authors:
Diab A.A.A.
Ref. 150 – Journal:
Are the authors feels certain with volume 11?
HSU C.-D.
Ref. 156 – Journal:
2009, 3, 332-336 - instead of 2009, “6”, 332-336
Ref. 157 – Author and Journal:
DÖTSCH J. – instead of “Dotsch” J.
2014, 1, 2
Ref. 170 – Journal.
2021, 2021, 5547228 – instead of 2021, “9”, 5547228
All comments were corrected as suggested.
We would like to thank Reviewer for time and effort reviewing this manuscript. I believe that the valuable hints will raised the value of this scientific publication. I hope that introduced changes solved the problem raised.
Round 2
Reviewer 4 Report
Comments:
P13-line 499:
“However, (comma) …”
Line 500:
“… of apelin compared to …”
Reference 136 (Kucur et al. 2014) is missing in the text!
Page 14 – line 552
In conclusion, the differences results between … needs rewriting i.e., “In conclusion, the different results between apelin/APF/ELABELA expression … may be explained by …”
Ref. 133 – No author listed – instead of no autor listed
Page 8 – first paragraph:
The reference of “Turco M.X.; Moffett, A. Development of the human placenta. Development 2019, 146, dev163428” - was deleted (Ref. 81 first version). Further references are therefore not valid because the authors did not change the text accordingly! In the former version, the authors mentioned: “ … at around 36-37 weeks of pregnancy [82]” - line 272. Reference 82 was cited by Proud J. 1987. The authors assigned the same passage in the current version to Ref. 81, i.e., Vaughan 2019 [former Ref. 83]. In this respect, it should be noted that Ref. 61 of the first version was a duplicate of Ref. 40, which was deleted in the present version. Therefore, the authors need to check each reference and revise the manuscript, accordingly.
Page 14 – line 549-51:
“In a very recent report, ELABELA reversed the phenotypes of PE in mice and regulated the expression of mouse placental apoptosis factors by reducing the levels of apoptotic genes [155].”
This reference (basic version 155) cites Ma J. et al. 2021. The same passage has now been given a different citation, i.e., Hamza et al. Int J Endocrinol 2021 (Ref. 154). In addition, Ref. 155 Simsek et al. was included but not cited properly, i.e., J Matern Fetal Med 2012, 25, 1705-1708.
Ref. 171 is new and incomplete:
Ho L., van Dijk M., Chye STJ, Messerschmidt DM, Chng SC, Ong S, Yi LK, Boussata S, Goh GH, Afink GB et al. ELABELA deficiency promotes preeclampsia and cardiovascular malformations in mice. Science 2017, 357, 707-713.
Author Response
We thank very much for all comments. Everything points has been corrected as suggested.